# Influence of Parameters on the Death Pathway of Gastric Cells Induced by Gold Nanosphere Mediated Phototherapy

**DOI:** 10.3390/nano12040646

**Published:** 2022-02-15

**Authors:** Jing Xin, Lei Fu, Jing Wang, Sijia Wang, Luwei Zhang, Zhenxi Zhang, Cuiping Yao

**Affiliations:** Key Laboratory of Biomedical Information Engineering of Ministry of Education, School of Life Science and Technology, Institute of Biomedical Photonics and Sensing, Xi’an Jiaotong University, Xi’an 710049, China; xinjing@mail.xjtu.edu.cn (J.X.); fulei2020@xjtu.edu.cn (L.F.); wangjing@mail.xjtu.edu.cn (J.W.); wang_sijia@mail.xjtu.edu.cn (S.W.); lwzhang@mail.xjtu.edu.cn (L.Z.)

**Keywords:** gold nanosphere-mediated phototherapy, nanosecond laser, photothermal effect, photochemical effect, vapor effect

## Abstract

Gold nanosphere (AuS) is a nanosized particle with inert, biocompatible, easily modified surface functionalization and adequate cell penetration ability. Photothermal, photochemical, and vapor effects of AuS could be activated by irradiating with nanosecond laser to cause cell death. Hence, AuS-mediated phototherapy irradiated with nanosecond laser is a promising and minimally-invasive treatment method for cancer therapy. However, various effects require different parameters to be activated. At present, few studies have reported on the influence of parameters of AuS inducing cell death under nanosecond laser irradiation. This makes it very challenging to optimize gold-nanoparticle-mediated specific or synergistic anti-cancer therapy. In this study, we revealed the main parameters and threshold values for AuS-mediated gastric cancer phototherapy with nanosecond pulsed laser irradiation, evaluated the pathway of induced cell death, and discussed the roles of photothermal, photochemical and vapor effects which can induce the cell death. The results showed that AuS-mediated phototherapy activated with nanosecond pulsed laser is an effective method for gastric therapy, mainly based on the photochemical effect. Prolonging the incubation time could decrease the irradiation dose, increase ROS-mediated photothermal effect and vapor effect, and then quickly induce cell death to improve security.

## 1. Introduction

Based on the ease of surface functionalization, uniquely tunable optical characteristics and photothermal heating ability, gold nanoparticles have been widely used in biosensors, cancer imaging contrast agents, photothermal therapy and drug delivery [1,2,3,4]. In particular, photothermal therapy (PTT), serving as a promising minimally-invasive treatment method, has a significant impact on treating benign disease and cancer [5]. This is because the quasi-free electrons in gold nanomaterials can interact with light and then undergo oscillation and ultimately induce surface plasmon resonance (SPR). The SPR can enhance the incident electric field in the vicinity of the gold nanoparticles. Localized elevation in temperature, acoustic wave signal and optical breakdown subsequently generated under laser irradiation. If the intensity of the laser is high enough, increasing heat is sufficient to damage cancer cells by releasing the intracellular constituents, which can cause a damaging inflammatory response into the extracellular components. Additionally, gold nanoparticles can be accumulated effectively inside cancer cells through their enhanced permeability and retention effect because of tumors of poor lymphatic drainage and abnormal vessel development.

However, nonthermal methods induced by gold nanoparticles could also result in cancer cell death under laser irradiation [6,7], such as reactive oxygen species (ROS)-dependent photochemical effect and vapor bubble formation based on the photomechanical effect, which can damage the cancer cells [8,9,10]. An increasing number of studies have indicated that ROS, which was widely generated by photosensitizers in photodynamic therapy to damage the cells, could be observed during the irradiation of metallic nanoparticles, especially gold nanoparticles [11,12]. Two distinct pathways can produce the ROS, albeit complementarily: a plasmon-activated pathway, which proceeds by the interactions of plasmons and hot electrons with molecular oxygen, and an indirect photothermal pathway, which induces extreme heat development causing particle fragmentation and increased thermionic electronic emission [8]. The generated ROS then triggers specific cell death mechanisms to damage cells [13]. Several studies have also indicated that the gold nanoparticles can produce transient vapor bubbles with irradiation at sufficiently high radiant exposures. The shear stress during vapor bubble expansion and collapse can mechanically damage the target cells [14]. The ROS-dependent photochemical effect and vapor bubble formation based on the photomechanical effect of gold nanoparticles with pulsed laser sources contribute significantly to cancer cell death during phototherapy. 

Gold nanospheres (AuS), as the most common shapes of gold nanoparticles, have a significant anticancer effect on ablating the cancer cells through PTT because they are inert, biocompatible, have easy modification of surface functionalization and have adequate cell penetration ability. The hyperthermal, vapor and photochemical effects generation induced by AuS under pulsed laser irradiation can be observed. These effects of gold nanoparticles under pulsed lasers are higher than those under continuous-wave lasers [15]. Compared with continuous wave lasers, nanosecond pulsed laser systems emit high-energy laser light in ultrashort pulse durations to induce AuS and generate the hyperthermia phenomenon, resulting in much more significant heat injury in PTT. Nano- or micro-scale transient vapor bubbles can be generated around AuS at high radiant energy coming from nanosecond pulsed laser systems [10]. The vapor bubble may cause cell death upon collapse. The ROS-dependent photochemical pathway is more pronounced in the case of AuS irradiation with pulsed laser than continuous wave laser [8]. In a word, a significant anti-cancer effect based on AuS has been shown and revealed that AuS is suitable for cancer therapy with pulsed laser irradiation. Presently, few studies have reported on the role of PTT, photochemical and vapor effects induced by AuS itself or AuS-nanoplatform irradiated by nanosecond laser on the cell damage under an appropriate condition, which can avoid toxicity on nearby healthy cells. This is because the conditions that induced these effects are different, and include irradiation dose, concentration, particle size, cell type and so on. 

Apoptosis and necrosis are common cancer cell death pathways after undergoing intense external stimuli, especially in gold nanoparticles-related cancer therapy [16,17,18]. As an unprogrammed cell death process, necrosis results in cytoplasmic swelling, organelle destruction, plasma membrane disruption, and then leads to intracellular content leak and consequent inflammation [1,14]. Otherwise, apoptosis is a programmed cell death form and is activated by proteases, causing cell shrinkage and nuclear fragmentation, as well as the appearance of apoptotic bodies with membrane blebs or blisters and small vesicles, and ultimately eradicating the cancer cell [16]. In PTT, cell death through necrosis is a relatively faster process than apoptosis, requiring somewhat higher intensity light irradiation and higher temperature (>50 °C) stimulation [19,20]. In typical heat damage procedures, gold nanoparticles-related PTT induces the cancer cell death mainly by necrosis as well [21]. In recent years, it has been reported that apoptosis can be activated in gold nanoparticles-related PTT to result in cell death by reducing light irradiation dose, increasing gold nanoparticle intracellular internalization efficiency, and so on [22,23,24,25]. The anti-cancer therapy through the apoptotic pathway to induce cell death can be used to reduce the side effects on normal cells. This is because apoptosis-inducing cell death can discourage serious inflammatory responses caused by high dose irradiation in PTT, and this apoptosis is more likely to be induced by the photochemical effect. In addition, it is recognized that low-energy irradiation induces apoptosis, and high-energy irradiation induces necrosis in gold nanoparticle-mediated phototherapy. Hence, the apoptotic pathway to induce cell death can be controlled by adjusting conditions such as the concentration of used gold nanoparticles and irradiation dose of light. This reveals the conditions that induced these effects of gold nanoparticles in the same induced cancer cell death event. Research into the condition and threshold value of gold nanoparticles, especially AuS, that result in apoptosis, such as the dose, the time of intracellular internalization, and the laser irradiation intensity, is essential to optimize the nanoparticle-mediated treatment system. 

Gastric cancer, as a common malignancy worldwide, has higher incidence and mortality in China than other countries. Cell specific responses related to toxicity and subsequent cell fate induced by AuS depend on the cell type [26]. Hence, revealing the main conditions and threshold value for AuS-mediated gastric cancer phototherapy with pulsed laser irradiation, clarifying the pathway of induced cell death under the conditions revealed above and discussing the roles of PTT, photochemical and vapor effects can greatly promote the application of AuS-mediated phototherapy in gastric cancer. In this study, the parameters that AuS resulted in gastric cancer cell death by apoptosis under nanosecond pulsed laser illumination was evaluated. To decrease dark cytotoxicity and increase stability, AuS was first modified by PEG polymer. PEG polymers can be used to increase biocompatibility, suppress immunogenic responses, and decrease adsorption to the negatively charged luminal surface of blood vessels. Therefore, the parameters and thresholds of PEG-coated gold nanospheres (PEG-AuS) under nanosecond pulsed laser radiation were demonstrated to induce cell death in gastric cancer from the concentration of PEG-AuS, treatment time, and the radiation energy of the nanosecond pulsed laser. Conditions of apoptosis-inducing occurrence and necrosis-inducing occurrence were then investigated. Finally, the photothermal, photochemical and vapor effects were also evaluated by temperature mapping detection, SOG and ROS concentration and optical scattering technique method. We found that the non-cytotoxic concentration of PEG-AuS in gastric cancer cells is 0.053 nM, and that concentration can be increased to 0.1075, or even to 0.215 nM if phototherapy is used, because 0.215 nM is the non-cytotoxicity concentration of PEG-AuS in normal gastric mucosal cells. Under treatment with 0.1075 nM AuS for 4 h, 22.93 mJ/cm^2^ radiation energy, apoptosis could be induced in gastric cancer cells and the SOG-mediated photochemical effect induced by AuS is better than its photothermal and vapor effect. After prolonging the treatment time for 6 h, even reducing the irradiation energy to 11.47 mJ/cm^2^ radiation energy, the gastric cancer cells were still able to induce death and the pathway of cell death mainly depended on necrosis. Under these conditions, the ROS-mediated photochemical effect induced by AuS is better than its photothermal and vapor effects (As shown in Figure 1). This revealed that prolonging the binding time of AuS can effectively improve the ROS-mediated photochemical effect to induce cell death and then decrease the irradiation dose to improve security while inducing cell death even by activating the cell necrosis. Generally, PEG-AuS under a nanosecond pulsed laser radiation is an effective agent and it is suitable for phototherapy and drug delivery systems for gastric cancer therapy. 

## 2. Material and Methods

### 2.1. Cell Lines and Reagents

The human moderately-differentiated gastric cancer cell line SGC-7901 purchased from the Cell Bank of the Academy of Military Medical Science (Beijing, China) and the human immortalized normal gastric mucosal cell line GES-1 obtained from the Beijing Institute of Cancer Research (Beijing, China) were donated by State Key Laboratory of Cancer Biology, the Digestion Department of XiJing Hospital. The SGC-7901 and GES-1 cells were cultured in RPMI1640 medium (HyClone) supplemented with 10% fetal bovine serum (HyClone) and 1% penicillin/streptomycina in a humidified incubator at 37 °C with 5% CO_2_. Cetyltrimethylamonium bromide (CTAB) was purchased from Sigma (St. Louis, MO, USA). Sodium borohydride (NaBH_4_), Chloroauric acid (HAucl_4_) and Ascorbic acid (AA) were purchased from Aladdin. A Cell Counting kit (CCK-8) was purchased from DoJindo (Kumamoto, Japan). The trypan blue, Hoechst 33258, PI Staining Kits, DCFH-DA fluorescence probe were purchased from Beyotime Company (Shanghai, China). Singlet oxygen sensor green reagent (SOSGR) was purchased from Sigma. SH-PEG-NHS was purchased from local supplier. 

### 2.2. Synthesis of PEG-AuS

The AuS was synthesized according to the seed-mediated growth method. Briefly, 0.01 M HAucl_4_ and 0.01 M cold NaBH_4_ in 0.1 M CTAB was standing reacted for 2 h at room temperature to yield small nanoparticles as the seed solution in which particle sizes are less than 5 nm. Then, 0.01 M HAucl_4_ and 0.1 M AA in 0.02 M CTAB was prepared as a growth solution. At last, 20 μL of the seed solution was introduced into the 24 mL of the growth solution and standing reacted. About 20 nm AuSs were obtained after 12–16 h. To remove excess CTAB capped at the surface of AuS, centrifugation was undertaken twice at 12,000 rpm. To reduce the toxicity of AuS, 200 μL 0.2 mM SH-PEG-NHS (MW 5000 Da) solution was added to 1 mL the AuSs solution at PEG:Au molar ratios of 1000. The mixtures were reacted for 24 h at room temperature on a rotary shaker and centrifuged twice at 8000 rpm for 15 min to remove the excess PEG reagent. Ultimately, PEG-AuSs were obtained and stored at 4 °C. The working solution of PEG-AuS was prepared by dilution of initial solution in fresh culture medium on the day of use.

### 2.3. Characterization of PEG-AuS

The absorption spectra of the AuS and PEG-AuS were measured by ultraviolet-visible spectrophotometer (V-550 UV/VIS, JASCO, Tokyo, Japan). The AuS and PEG-AuS morphologies were obtained using transmission electron microscopy (TEM) (JEM-2100, JEOL, Tokyo, Japan). Dynamic light scattering (DLS) and the zeta potential of AuS and PEG-AuS were observed with a Malvern Zetasizer Nano ZS (ZS90, Malvern, UK). 

### 2.4. Cytotoxicity Assay

Cell viability assay to evaluate dark cytotoxicity of PEG-AuS on gastric cancer cells and immortalized normal gastric mucosal cells were measured by CCK-8 assay. Briefly, SGC-7901 and GES-1 cells (2.5 × 10^4^ cells/mL) were seeded to sterile 96-well flat-bottomed plates, and incubated overnight. Diluted PEG-AuSs were then added to each well with the final concentration from 0.0134 nM to 1.72 nM based on the 2-times step dilution method. In this 96-well flat-bottomed plate, three wells containing only cells were used as control, and three wells containing only complete culture medium were used as blank control. The plates were then incubated at a humidified incubator at 37 °C with 5% CO_2_ for 2 h, 4 h, 6 h, respectively. After the incubation, the medium contained PEG-AuSs were replaced by fresh 1640 culture medium. The plates were then continuously incubated for 24 h. After incubation, the solution containing 100 μL 1640 medium and 10 μL CCK-8 was added to each well. The plates were then incubated for 1 h in a humidified incubator at 37 °C with 5% CO_2_. Finally, the plates were measured for absorbance levels at 450 nm using a microplate reader (Infinite M200 Pro., Tecan, Männedorf, Switzerland). The dark cytotoxicity of PEG-AuSs with different concentration and different incubated time was calculated as ((OD of PEG-AuSs treated-OD of blank control)/(OD of control-OD of blank control)) × 100%. 

### 2.5. Trypan Blue Staining Assay

Trypan blue staining assay to evaluate immediate cell damage effects of PEG-AuSs on gastric cancer cells and immortalized normal gastric mucosal cells were measured. 2.5 × 10^5^ SGC-7901 cells and GES-1 cells were harvested at logarithmic growth phase and grown in six well plates and allowed to attach overnight. Then, based on the principle of less cytotoxicity on normal gastric tissue cells and high photocytotoxicity effect on gastric cancer cells, 0.1075 nM PEG-AuSs were used to pre-treat cells for 4 h and 6 h, respectively, at a humidified incubator at 37 °C with 5% CO_2_. After the incubation, the plates containing PEG-AuSs were washed with PBS twice and the medium replaced by fresh 1640 culture medium. The cells were then irradiated by the 532 nm nanosecond pulsed laser systems for 5 min with different radiant exposures (from 5.74 mJ/cm^2^ to 91.72 mJ/cm^2^). The radiant intensity of nanosecond pulsed laser was from 1.5 to 24 mJ, the actual duration of the nanosecond pulsed laser was 0.1 s, the radius of the laser beam was 0.5 cm. Hence, the radiant dose in this study was from 5.74 mJ/cm^2^ to 91.72 mJ/cm^2^. The direct cell damage is tested by 0.4% trypan blue staining for 2 min. Dead cells can be immediately absorbed by the staining dye and stained blue, while living cells cannot accumulate the staining dye and stained blue. After staining, the cells are imaged under 10× in bright field using a Nikon eclipse Ti fluorescence microscope.

### 2.6. Photocytotoxicity Assay

After revealing the safe concentration and radiation dose of PEG-AuS, the photocytotoxicity to induce cell death was evaluated. The cells were treated in the same manner as described previously. However, after the medium containing PEG-AuSs was replaced by fresh 1640 culture medium, the cells should be irradiated by a nanosecond pulsed laser systems (Q-smart 450, Quantal, France) for 5 min with different radiant exposures (11.47 mJ/cm^2^ and 22.93 mJ/cm^2^). Then, the plates were incubated at a humidified incubator at 37 °C with 5% CO_2_ for 24 h. The photocytotoxicity effect analysis using CCK-8 was completed in a similar manner as the cytotoxicity assay described above.

### 2.7. Detection of Cell Apoptosis and Necrosis 

The nuclei morphology and different fluorescence stains caused by photocytotoxicity of PEG-AuSs were visualized by Hoechst 33324/Propidium Iodide (PI) nuclear staining kits according to the manufacturer’s instructions. Briefly, SGC-7901 cells and GES-1 cells (2.5 × 10^5^ cells/mL) were seeded on bio-clean cover slips in 24-well plates and allowed to attach overnight. Then, the cells were treated with 0.1075 nM, 0.0537 nM, 0.0269 nM, 0.0134 nM, 0.0067 nM PEG-AuS or fresh complete culture medium for different time span (4 h and 6 h) at a humidified incubator at 37 °C with 5% CO_2_. After the incubation, the plates were washed twice with PBS and the medium replaced by fresh 1640 culture medium. The cells were then irradiated by the 532 nm nanosecond pulsed laser systems for 5 min with different radiant exposures (11.47 mJ/cm^2^ and 22.93 mJ/cm^2^) and then incubated in a humidified incubator at 37 °C with 5% CO_2_ for 1 h or 24 h again. After treatment, the cells were stained with Hoechst for 10 min and PI for 5 min. Cover slips were then washed and mounted on slides with glycerol and imaged with fluorescence microscope before being described. To quantify the percentage of apoptosis, we counted the number of cells with apoptotic and necrosis characteristics among 200 cells at high power field. Flow cytometry is another powerful technology for the analysis of apoptosis and necrosis, which quantifies phosphatidylserine exposure on the surface of apoptotic cells using Annexin-V FITC and PI stain. The cells were treated with PEG-AuSs and irradiated by the 532 nm nanosecond pulsed laser systems according to the same treatment conditions as above. The cells were then harvested, centrifuged at 800 rpm for 5 min, washed with PBS, centrifuged again, resuspended in PBS, and stained with Annexin-V FITC for 1 h and PI for 30 min, and finally analyzed by a FACScan system.

### 2.8. SOG and ROS Generation

In gold nanoparticle-mediated phototherapy, ROS and SOG are important factors to induce photochemical effect. Therefore, the generation of ROS and SOG were evaluated after treatment with different concentrations of PEG-AuSs (from 0.0067 nm to 0.1075 nm) at different radiant exposures (11.47 mJ/cm^2^ and 22.93 mJ/cm^2^). After irradiation, the cells were washed twice with PBS and harvested and incubated with 10 μmol/L DCFH-DA for 20 min at 37 °C in complete darkness, PBS washed, and then detected by a fluorescence spectrophotometer under the excitation of 488 nm light for ROS detection. Or, the cells were washed, harvested permeabilized with 0.5% Triton X-100 in PBS for 10 min, centrifuged, washed with PBS, mixed with SOSGR probe, irradiated with 635 nm laser system for 5 min, and then measured by a fluorescence spectrophotometer under excitation of 504 nm light for SOG detection.

### 2.9. Temperature Mapping Detection

To understand the photothermal effect of PEG-AuSs induced cell death, the temperature mapping image representing the photo-heat transition were recorded with a thermal imaging system (Lynx GigE, Xenics) during laser illumination. The cells were treated with PEG-AuSs at different concentrations for 4 h and 6 h. Then the cells were washed with PBS and placed at the central focus of the 532 nm pulse beam and the focusing lens of this thermal imaging system. The illumination time is 5 min.

### 2.10. Vapor Bubbles Detection

The vapor bubbles formation around the gold nanoparticles is an important phenomenon to characterize the optical breakdown induced by laser pulses and can also cause damage to the targeted cells. Hence, whether the formation of vapor bubbles around PEG-AuSs occurs or not during our experiment for the high peak intensity of short duration laser pulses should be evaluated. Two different kinds of optical scattering technique methods were introduced to detect the vapor bubbles around PEG-AuSs induced by laser pulses. The vapor bubbles detection platform was built according to the schematic diagram of the designed optics system. Pulse beams generated by a 532 nm frequency-doubled, Q-switched Nd:YAG laser (Q-smart 450, Quantal, France) and CW beams generated by a 660 nm continuous wave laser (OBIS, Coherent, Santa Clara, CA, USA) were focused in the liquid of the cuvette by an objective (40×, 0.6). Bubbles at the central focus nearby, caused by pulse laser scatters of the CW beam, increasingly reduce the axial intensity of the CW beam during its expansion stage, and brings it back to the base level during its collapse stage. The change in axial intensity of the CW beam represented the vapor bubble formation, and was detected by a high-speed photodetector (HAS-X-S-1 G4-S, FEMTO, Berlin, Germany). Meanwhile, an EMCCD (ixon3, Andor, London, England) at the side of the cuvette was used to obtain the scattering imaGES-1 of bubbles. The vapor bubbles around the gold nanoparticles can strongly enhance the side scattering properties of gold nanoparticles, the intensity of light scattered from laser pulses can be significantly enhanced for the bubbles generated during moments of laser pulse duration. 

### 2.11. TEM Observation 

TEM analysis of the sliced gastric cell and immortalized normal gastric mucosal cell was done to provide the cellular uptake and intercellular distribution of PEG-AuSs. 2.5 × 10^5^ SGC-7901 cells and GES-1 cells were seeded in 6-well plates and cultured overnight. Then 0.1075 nM PEG-AuSs were treated into the cells for 4 h or 6 h. After being incubated and washed three times with PBS, the cells were fixed with 2.5% glutaraldehyde/0.1 M PBS (PH = 7.4) overnight at 4 °C, washed with PBS, fixed with 1% osmium tetroxide and 1.5% potassium ferrocyanide for 1.5 h at room temperature, washed with PBS, and then fixed with 2.5% glutaraldehyde/0.1 M PBS (PH = 7.4) overnight at 4 °C again. After being fixed, the cells were dehydrated in a graded ethanol series and embedded in Epon 812 for 3 h. Ultrathin sections were then cut on an RMC Ultrmicrotome MTX. At last, the sections were stained with 2% uranyl acetate for 15 min followed by lead citrate for 10 min and examined by TEM.

### 2.12. Statistical Analyses

Data in this study are presented as the mean ± S.D. of three replicate experiments. All statistical analyses were done by SPSS18.0 software (Chicago, IL, USA). Statistical difference between the means was analyzed with Student’s t test. Significance was set at the 5% level.

## 3. Results and Discussion

### 3.1. Synthesis and Characterization of PEG-AuS

As indicated in Figure 2, the diameter of AuS synthesized using the seed-mediated growth method is approximately 15 nm, and the maximum surface plasmon absorption is at 524 nm. The diameter of PEG-AuS is changed to 21 nm and the maximum surface plasmon absorption is changed to 531 nm. The hydrodynamic size and zeta potential of AuS and PEG-AuS are 27.9 nm, 27.4 mV and 61.8 nm, 1.37 mV, respectively. Hence, the PEG polymer changed the size and zeta potential of naked AuS, which decreases the uptake of the reticuloendothelial system and prolongs the circulation time [27].

### 3.2. Cytotoxicity of PEG-AuS

To determine the initial value of PEG-AuS induced cytotoxicity on SGC-7901 gastric cancer cell line and GES-1 immortalized normal gastric mucosal cell line, CCK-8 cell viability analysis was performed. The incubation time and the dose of PEG-AuS are essential factors that influence the cytotoxicity. Hence, the cell viability of PEG-AuS was evaluated with the incubation time or the change in dose. As shown in Figure 3, 0.0537 nM PEG-AuS induce little cytotoxicity on SGC-7901 cells, despite the incubation time being 2, 4 and 6 h. With an increase in the dose to 0.1075 nM, the cell viability at 2, 4 and 6 h was decreased to 86%, 66%, and 58% respectively. Furthermore, when the dose of PEG-AuS or the incubation time was increased, the cell viability decreased accordingly. Increasing the dose to 0.43 nM, the cell viability was nearly inhibited at 6 h, and increasing the dose to 1.72 nM, the cell viability was completely inhibited even at 2 h. Compared with SGC-7901 cells, the low cytotoxicity of PEG-AuSs on GES-1 cells was below 0.215 nM. Above 0.43 nM, the cell viability was quickly reduced, regardless of the incubation time (2, 4 and 6 h). Generally, the incubation time and the drug dose had a higher influence on SGC-7901 cells compared with normal GES-1 cells. This may be caused by its higher intracellular uptake efficiency related with proliferation ability. Hence, the used concentration of PEG-AuSs for drug delivery and imaging diagnose should under 0.0537 nM for 6 h as much as possible and no more than 0.1075 nM for 4 h, and based on the principle of killing tumor cells to the greatest extent while having little toxic effect on normal cells to gastric cancer therapy, the concentration can be increased to 0.215 nM for 6 h and even to 0.43 nM for 2 h.

### 3.3. Immediate Cell Damage Value of Irradiation Dosage Based on the Nanosecond Pulsed Laser Systems

In this study, the research emphasizes the anti-tumor effect of PEG-AuSs after irradiation by the nanosecond pulsed laser systems. Obviously, the anti-tumor effect is related to the irradiation dosage. This is because nanosecond pulsed laser systems have high-energy laser light in ultrashort pulse durations, and high irradiation dosage can be immediately induced to cell damage. Hence, it is essential to test the immediate cell damage values of irradiation dosage based on the nanosecond pulsed laser systems to decrease the destruction of normal cells. To find the immediate cell damage values of irradiation dosage, the trypan blue staining assay was evaluated. The bright field transmission images of cells stained with trypan blue dye are shown in Figure 4. At 4 h incubation time, 0.1075 nM concentration and 45.86 mJ/cm^2^ radiation dosage, both SGC-7901 cells and GES-1 cells were destroyed. Under 45.86 mJ/cm^2^, the cells were not affected immediately. Therefore, the immediate cell damage values of the irradiation dosage of PEG-AuSs irradiated by the nanosecond pulsed laser systems were 45.86 mJ/cm^2^ at 4 h treatment time and there was no significant difference between SGC-7901 and GES-1 cells. However, changing the incubation time to 6 h, the radiation dosage was obviously down to damage the cell immediately. The values had a significant difference between SGC-7901 and GES-1 cells. Even at 11.47 mJ/cm^2^, a fraction of SGC-7901 cells still exhibited damage characteristics. Reducing the irradiation dosage to 5.47 mJ/cm^2^, the phenomenon of damage disappeared. Hence, the initial values of irradiation doses to induce the cell damage immediately on SGC-7901 cells for 6 h was 5.74 mJ/cm^2^. Compared with the SGC-7901 cells, the initial values of irradiation doses to induce cell damage immediately on GES-1 cells for 6 h increased to 45.86 mJ/cm^2^. At 11.47 mJ/cm^2^ or 22.93 mJ/cm^2^, few cells of GES-1 were damaged. In addition, the cell debris observed from bright field transmission images of GES-1 indicated that when exposed to the high-power dosage of 91.72 mJ/cm^2^, the cell cytoplasm membrane and cell nucleus membrane can be destroyed. Besides the hyperthermia phenomenon induced by PEG-AuSs, the mechanical stress that comes from the high-power pulsed laser may also lead to the rupture and fragmentation of the cells. Therefore, the used irradiation dose of PEG-AuSs for drug delivery and imaging diagnosis should be under 45.86 mJ/cm^2^ for 4 h and 11.47 mJ/cm^2^ for 6 h as much as possible, and it can be increased to 22.93 mJ/cm^2^ for 6 h for phototherapy. 

### 3.4. Antigrowth Effect of PEG-AuS Irradiated by the Nanosecond Pulsed Laser System

In this study, based on the principle of little cytotoxicity on normal gastric tissue cells and high photocytotoxicity effect on gastric cancer cells, the antigrowth effect of PEG-AuSs were further investigated with different concentrations from 0.0067 nM to 0.1075 nM and different incubation times (4 h or 6 h) irradiated by the nanosecond pulsed laser systems on SGC-7901 and GES-1 cell lines. Based on the obtained immediate cell damage threshold, the used irradiation dosage for anti-tumor therapy on gastric cancer is 11.47 mJ/cm^2^ or 22.93 mJ/cm^2^. The cell survival rates are given in Figure 5. The results indicated that PEG-AuSs had a significant inhibition effect on SGC-7901 cells compared with GES-1 cells. The antigrowth effectiveness of PEG-AuSs can be affected by the concentration of the agent, the incubation time and radiation dose. At 0.0537 nM for 4 h, the cell survival rate of SGC-7901 cells was approximately 72% at 11.47 mJ/cm^2^ and approximately 34% at 22.93 mJ/cm^2^. Increasing the incubation time to 6 h, the cell survival rate of SGC-7901 cells decreased to approximately 28% at 11.47 mJ/cm^2^ and approximately 26% at 22.93 mJ/cm^2^. When the concentration was altered to 0.1075 nM, the cell survival rate of SGC-7901 cells decreased to about 52% at 11.47 mJ/cm^2^ and about 16% at 22.93 mJ/cm^2^. Increasing the incubation time to 6 h, the cell survival rate of SGC-7901 cells was further reduced to about 12% at 11.47 mJ/cm^2^ and about 8% at 22.93 mJ/cm^2^. Compared with SGC-7901 cells, GES-1 exhibited a wake antigrowth effect at the same treatment conditions. Even at the harshest condition in this study (the drug concentration: 0.1075 nM, the irradiation dosage: 22.93 mJ/cm^2^ and the incubation time: 6 h), the cell survival rate of GES-1 can still reach approximately 70%. Therefore, PEG-AuSs had a significant anti-growth effect on gastric cancer cells irradiated by the nanosecond pulsed laser systems at 0.1075 nM, 22.93 mJ/cm^2^ for 4 h or at 0.1075 nM, 11.47 mJ/cm^2^ for 6 h. The used radiation dose to induce cell death can be decreased through increasing the incubation time. 

### 3.5. Apoptosis and Necrosis Induced by PEG-AuS Irradiated by the Nanosecond Pulsed Laser Systems

The significant antigrowth effect can be obtained at 6 h incubation time and 11.47 mJ/cm^2^ irradiation dosage or at 4 h incubation time and 22.93 mJ/cm^2^ irradiation dosage. Is there a difference in the method of cell death induced by them? Hence, the apoptosis and necrosis were further evaluated at 0.1075 nM, 0.0537 nM, 0.0269 nM, 0.0134 nM, 0.0067 nM by Hoechst 33324/PI nuclear staining kits. Cell shrinkage, nuclear fragmentation and so on as cell emblematic apoptotic characteristics can be observed with Hoechst 33,324 staining dye and cell plasma membrane unintegrated as cell representative necrosis characteristics can be stained with PI staining dye (PI cannot cross the membrane of live cells). Therefore, it is an effective method to distinguish apoptosis and necrosis. In this study, the cells showed significant difference in apoptosis and necrosis after being treated under different conditions, as shown in Figure 6 and Appendix A. Specifically, apoptosis was the main method of death after irradiation at 11.47 mJ/cm^2^ for 4 h, and the percentage of apoptosis depended on the concentration of PEG-AuSs. With an increased laser irradiation dosage to 22.93 mJ/cm^2^, the necrosis situation was significantly increased at 0.1075 nM. However, the major of that was still apoptosis. In contrast, the necrosis was the main form of death after irradiation at 11.47 mJ/cm^2^ for 6 h at 0.0537 and 0.1075 nM, and the necrosis phenomenon can be contained quickly. With an increase in the laser irradiation dosage to 22.93 mJ/cm^2^, necrosis was quickly induced within 1 h and then the apoptosis was still induced to lead to the other cells death at 0.0269, 0.0537 and 0.1075 nM. It revealed that higher concentration PEG-AuS internalized into the cells through prolonged incubation time can increase the necrosis-inducing ability in a short time. Low laser irradiation dosage can also induce cell death in a short time by increasing the incubation time. It may be caused by more PEG-modified gold nanospheres internalized in the cells.

### 3.6. Cellular Uptake of PEG-AuS

To further determine the reason for the different antigrowth effects and pathways of cell death of the PEG-AuSs with different incubation times (4 h and 6 h), the cellular uptake ability and location of internalized were observed by TEM. As shown in Figure 7, the intracellular distribution of PEG-AuSs in SGC-7901 cells with 4 h and 6 h incubation time was mainly in the cytosol and lysosome, respectively. Compared with SGC-7901 cells, the intracellular distribution of PEG-AuSs in GES-1 cells with 4 h and 6 h incubation time were mainly in the vesicle. The TEM results also showed that the PEG-AuSs particle uptake into the SGC-7901 cells through transcytosis and endocytosis, resulting in PEG-AuSs particles located freely in the cytosol and then located in the lysosome with the time prolonged. Compared with the SGC-7901 cells, pinocytosis may be the main route for the uptake into GES-1 cells, resulting in PEG-AuSs particles located in the vesicle. Additionally, the agglomeration of PEG-AuSs in GES-1 cells was higher than that of SGC-7901 cells. PEG-AuSs uptake into the GES-1 cells may be induced through mycropincytosis, and ultimately localized in the vesicle. The agglomeration of PEG-AuSs resulted in diminished uptake into the GES-1 cells and weak cytotoxicity and phototoxicity on GES-1 cells. 

### 3.7. Change of Temperature Induced by PEG-AuS under Nanosecond Pulsed Laser Irradiation

In PTT, the temperature should be greater than 43 °C or the increased temperature should be higher than 6 °C, which can only cause ablation of tumor tissue through protein denaturation and disruption of the cellular membrane [28,29,30]. Hence, to confirm whether the hyperthermic effect is the reason for cell damage by PEG-AuS under nanosecond pulsed laser irradiation at the used exposure dose or not, the temperature of treated cells should be evaluated. As shown in Figure 8, the hyperthermic effect can be induced by PEG-modified gold nanospheres themselves, even at 11.47 mJ/cm^2^ exposure. Under 11.47 mJ/cm^2^ and 22.93 mJ/cm^2^ irradiation dose, the temperature was raised by 6 °C and 10 °C, respectively, in 5 min. However, the temperature was raised slightly after they entered the cells. At 4 h incubation time, only 1 °C and 2 °C was raised after 11.47 mJ/cm^2^ and 22.93 mJ/cm^2^ irradiation. At 6 h incubation time, only 2 °C and 3 °C were raised after 11.47 mJ/cm^2^ and 22.93 mJ/cm^2^ irradiation. Increasing the exposure dose to 45.86 mJ/cm^2^, the temperature was raised by 4 °C. The increased temperature was limited, which may be caused by the lower concentration of PEG-modified gold nanospheres in the cells. Regardless, at the used irradiation dose, the temperature cannot reach the value to induce the hyperthermic effect. Hence, the PTT effect may not be the immediate factor to induce cell damage at the range of the used irradiation dose and concentration.

### 3.8. Photochemical Effect Based on ROS or SOG Generation Induced by PEG-AuS under Nanosecond Pulsed Laser Irradiation

The primary mechanism of photochemical therapy (no matter photodynamic therapy or photothermal therapy) damages cellular organelles and membranes and induces cell death. After light radiation, the material could generate ROS, in particular SOG, to induce oxidative stress and then damage the cell structure. Therefore, the photochemical effect induced by PEG-AuSs can be revealed by measuring the generation of ROS and SOG. As indicated in Figure 9, under the condition of the used concentration of PEG-AuSs (0.1075 nM, 0.0537 nM, 0.0269 nM, 0.0134 nM, 0.0067 nM), the concentration of SOG and ROS were also elevated. In addition, the results showed that ROS can be more easily induced with the incubation time prolonged, and the irradiation dose and gold nanoparticle concentration lowered. On the contrary, SOG could be easily induced with an irradiation dose even at a low concentration of PEG-AuSs. By general comparison, the photochemical effect may be the main factor to induce the cell damage at the range of the used irradiation dose, concentration, and incubated time. Under treatment with 0.1075 nM PEG-AuSs for 4 h, 22.93 mJ/cm^2^ radiation energy and 5 min exposure time, the ROS-mediated photochemical effect is the main effect to induce cell death, and under treatment with 0.1075 nM PEG-AuSs for 6 h, 11.47 mJ/cm^2^ radiation energy and 5 min exposure time, SOG-mediated photochemical effect is the main effect to induce cell death.

### 3.9. Threshold of Irradiation Dosage to Induce Vapor Bubble Based on the Nanosecond Pulsed Laser System

After pulsed laser irradiation, the local temperature of AuS rapidly rises. If the maximum temperature achieved in the AuS is sufficiently high, it vaporizes the surrounding liquid and results in the generation of a vapor bubble with the minor size of nanoscale and minor lifetime of nanosecond scale. Laser induced vapor bubbles around AuS can break the cell membrane easily and hence haves been widely used in cell operation and cancer therapy [31,32,33,34]. Hence, we further investigated whether the vapor bubble was generated or not during laser pulse irradiation under the conditions of the used concentration of PEG-AuSs. It is difficult to directly measure the dynamics of such a small transient vapor bubble for its small size and short lifetime. However, the generation of a vapor bubble around a nanoparticle results in a drastic change in optical scattering properties, which is widely introduced to detect vapor bubbles [35,36,37,38,39,40]. When a vapor bubble is generated, the intensity of side-scattering light is enhanced but the intensity of forward-scattering light is reduced. Based on this, the experimental setup depicted in Figure 10A was built to detect laser induced vapor bubbles around PEG-AuSs through side-scattering imaging and forward-scattering detection. A pulsed laser is focused into a quartz cuvette (inner size 10 mm × 10 mm, wall thickness 1 mm) that is filled with AuS solutions by a long–working distance objective with a 10×, 0.6 numerical aperture (Daheng optics, GCO–2133). A continuous expanded probe beam emitted from an He–Ne laser (Thorlabs, HNL020 RB) is adjusted collinear with the pulsed laser beam, and focused into the cuvette. If a vapor bubble is generated, due to the enhancement of side scattering property, more of the exciting light is scattered and hence can be captured by the camera, as Figure 10B (top) shows. Besides, the oscillation of vapor bubbles also results in the change in the intensity of the forward probe beam, which can be detected by a photodetector, as Figure 10B (bottom) shows. According to this, the bubble lifetime can be measured and then its size can be calculated. However, the side–scattering imaging technique is more sensitive, which can detect smaller sized bubbles. Figure 10C evaluated the probability of vapor bubble generation as the function of laser fluence for PEG-AuSs solutions, the dash line is the logistic fit curve of experimental data. The fluence threshold of vapor bubbles for 10% probability is 39.4 mJ/cm^2^ and for 50% probability is 60.9 mJ/cm^2^ by side-scattering imaging, which is much smaller than the fluence threshold for 10% probability of 100.0 mJ/cm^2^ and for 50% probability of 329.8 mJ/cm^2^ by forward scattering detection. It revealed that side-scattering imaging has higher sensitivity for the detection of small vapor bubbles. The results also showed that the fluence threshold of the vapor bubble is larger than the fluence of the unfocused laser pulse (11.47 mJ/cm^2^ and 22.93 mJ/cm^2^) that is used for cancer cell damage. It seems that vapor bubbles are barely generated during the PEG-AuS-induced cell damage experiment. However, the inevitable agglomeration of intracellular AuSs strongly increase its absorption cross-section, and then decrease the fluence threshold of vapor bubble generation. Hence, we also investigate the influence of agglomeration of PEG-AuSs on fluence threshold of vapor bubble. The absorption spectra and TEM images of PEG-AuSs aggregates was shown in Appendix A. Only the side–scattering imaging method was used to evaluate the influence of agglomeration of PEG-AuSs. The results are shown in Figure 10D. Apparently, with the increase in agglomeration degree of PEG-AuSs, the vapor bubble threshold is significantly reduced. For PEG-AuS–cluster solutions, the fluence threshold for 10% probability is mostly below 10 mJ/cm^2^. This indicates that vapor bubbles could be formed when a laser pulse with fluence of 11.47 mJ/cm^2^ is used for the treated time of 6 h. Hence, the generation of a vapor bubble around AuSs may also contribute to the death of the cell for its strong localized mechanical effect. 

## 4. Conclusions

In this study, we revealed the main parameters and threshold values for PEG-AuS-mediated gastric cancer phototherapy with nanosecond pulsed laser irradiation, evaluated the pathway of induced cell death under the conditions revealed above and discussed the roles of PTT, photochemical and vapor effects which can induce the cell death. The results showed that the non-cytotoxic concentration of PEG-AuS in gastric cancer cells is 0.053 nM, therefore the used concentration of PEG-AuSs for drug delivery and imaging diagnosis should be under 0.0537 nM for 6 h as much as possible. In addition, based on the principle of killing tumor cells to the greatest extent while having little toxicity on normal cells in gastric cancer therapy, the concentration can be increased to 0.1075, or even to 0.215 nM if used in phototherapy, because 0.215 nM is the non-cytotoxic concentration of PEG-AuS in normal gastric mucosal cells. Treated with 0.1075 nM PEG-AuS for 4 h, 22.93 mJ/cm^2^ radiation energy (6 mJ radiant intensity, 3000 pulses, 0.5 cm radiation radius) or 6 h, 11.47 mJ/cm^2^ radiation energy (3 mJ radiant intensity, 3000 pulses, 0.5 cm radiation radius), the anti-growth effect could be exhibited significantly. Treated with 0.1075 nM PEG-AuS for 4 h and 22.93 mJ/cm^2^ radiation energy, apoptosis can be induced in gastric cancer cells and the SOG-mediated photochemical effect induced by AuS is better than its photothermal and vapor effects. After prolonging the treated time to 6 h, even reducing the irradiation energy to 11.47 mJ/cm^2^, cell death could be still induced in the gastric cancer cells and the pathway of cell death mainly depended on necrosis. Under these conditions, the ROS-mediated photochemical effect induced by AuS is better than its photothermal and vapor effects. The threshold of irradiation dosage to induce vapor effect could be decreased by AuS aggregation. This revealed that PEG-AuS could induce cell death quickly through ROS-mediated photochemical effect and vapor effect with low irradiation by prolonging the binding time of AuS. Generally, PEG-AuS under nanosecond pulsed laser radiation is a safe and effective agent and it is suitable for phototherapy and drug delivery systems for gastric cancer therapy. 

## Figures and Tables

**Figure 1 nanomaterials-12-00646-f001:**
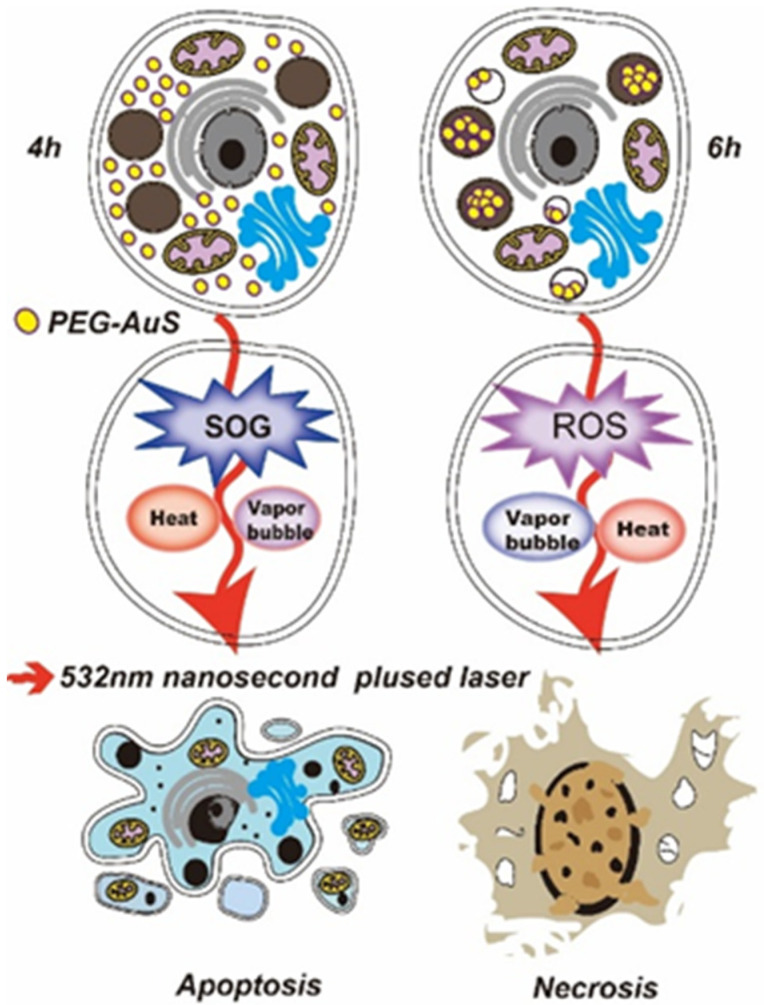
The effect of PEG-coated gold nanospheres (PEG-AuS) for phototherapy under nanosecond pulsed laser radiation.

**Figure 2 nanomaterials-12-00646-f002:**
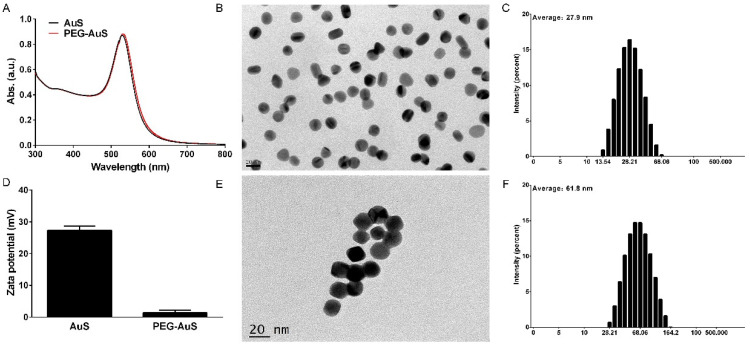
Synthesis and properties of AuS and PEG-AuS. Notes: (**A**) UV-vis absorption spectra of AuS and PEG-AuS. (**B**,**E**) Transmission electron microscopy images of AuS and PEG-AuS. (**C**,**F**) Hydrodynamic size distribution of AuS and PEG-AuS. (**D**) Zeta potential of AuS and PEG-AuS.

**Figure 3 nanomaterials-12-00646-f003:**
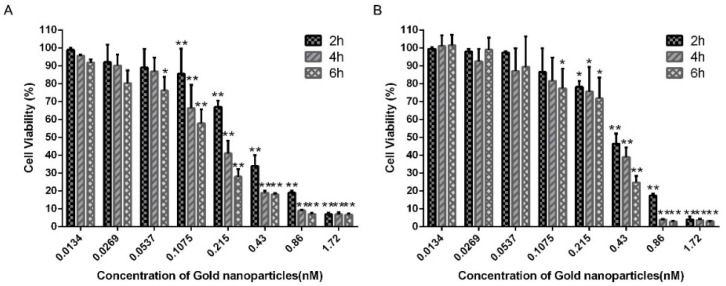
The dark cytotoxicity of PEG-AuS. Notes: (**A**) The cytotoxicity of PEG-AuS on SGC-7901 cells. (**B**) The cytotoxicity of PEG-AuS on GES-1 cells. *: *p* < 0.05; **: *p* < 0.01.

**Figure 4 nanomaterials-12-00646-f004:**
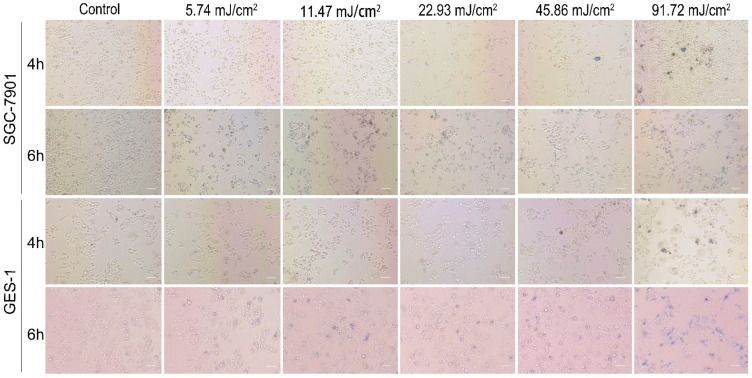
The Trypan blue staining of SGC-7901 cells and GES-1 cells after treated with PEG-AuS irradiated with different irradiation dose.

**Figure 5 nanomaterials-12-00646-f005:**
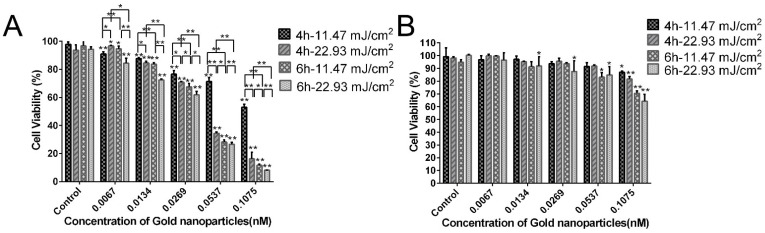
The anti-growth effect of PEG-AuS on SGC-7901 cells (**A**) and GES-1 cells (**B**) after treated with PEG-AuS with different conditions by CCK-8 assay. *: *p* < 0.05; **: *p* < 0.01.

**Figure 6 nanomaterials-12-00646-f006:**
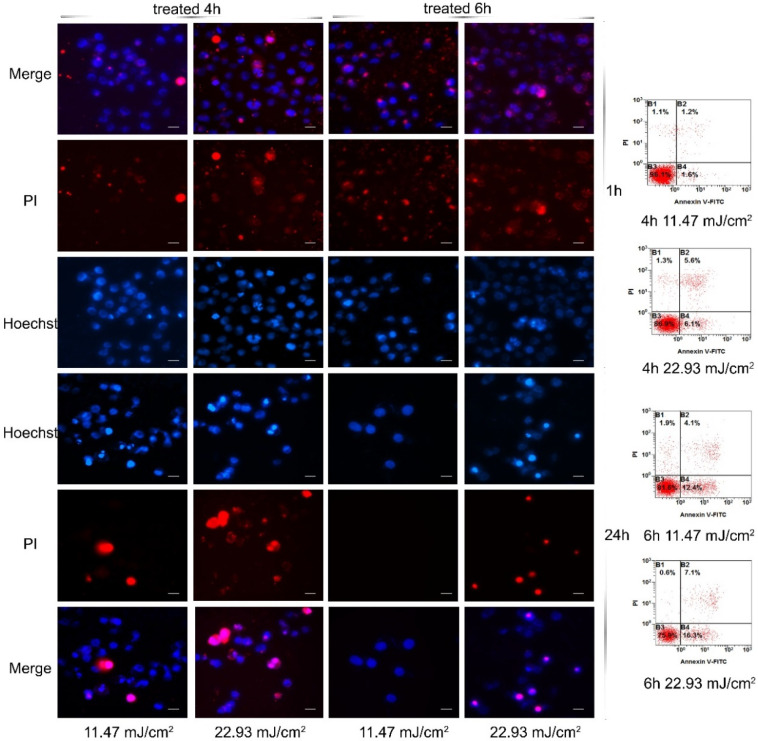
Apoptosis and necrosis induced by PEG-AuS irradiated with nanosecond laser.

**Figure 7 nanomaterials-12-00646-f007:**
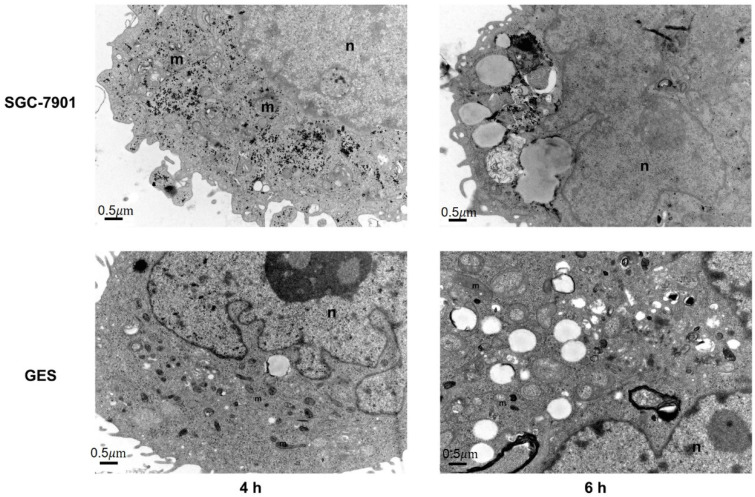
The transmission electron microscopy images of PEG-AuS on SGC-7901 and GES-1 cells after treatment with PEG-AuS for 4 h or 6 h.

**Figure 8 nanomaterials-12-00646-f008:**
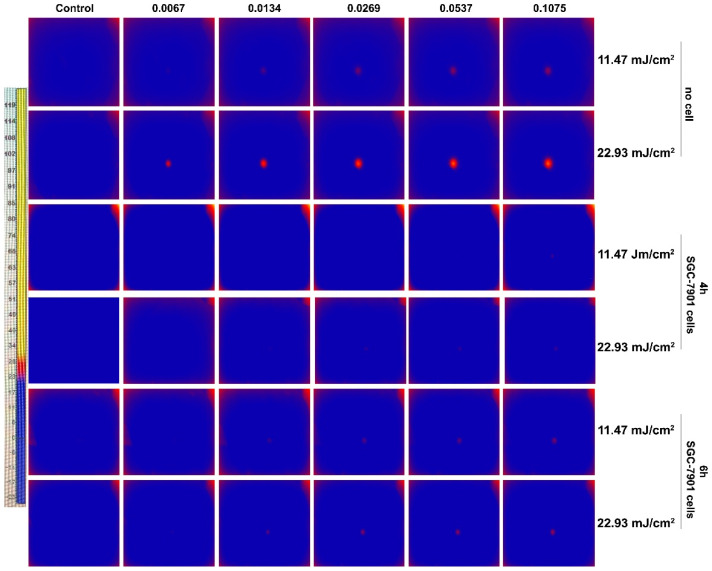
The changed temperature induced by PEG-AuS measured by thermal imaging system.

**Figure 9 nanomaterials-12-00646-f009:**
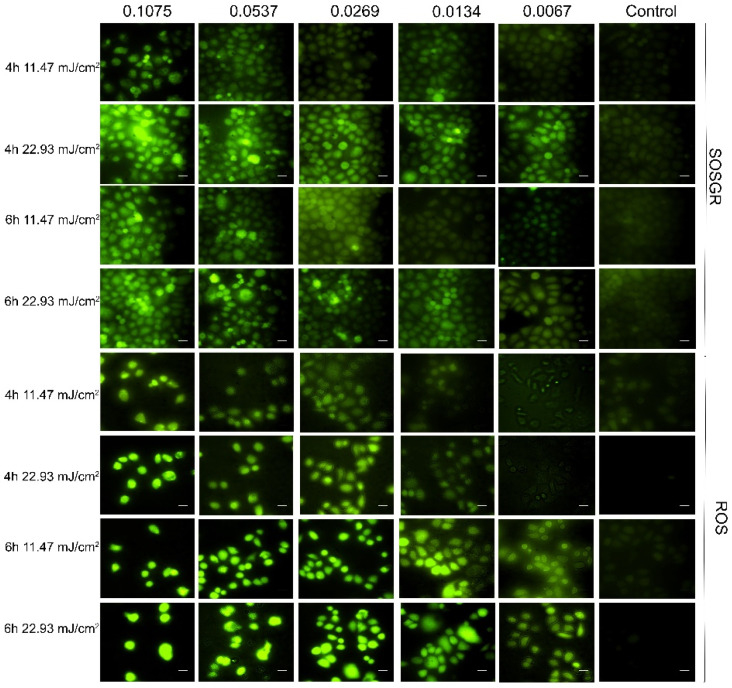
SOG and ROS production induced by PEG-AuS at different concentration and different incubation time (4 h or 6 h) and irradiated with nanosecond pulsed laser with different irradiation dose (11.47 mJ/cm^2^ and 22.93 mJ/cm^2^).

**Figure 10 nanomaterials-12-00646-f010:**
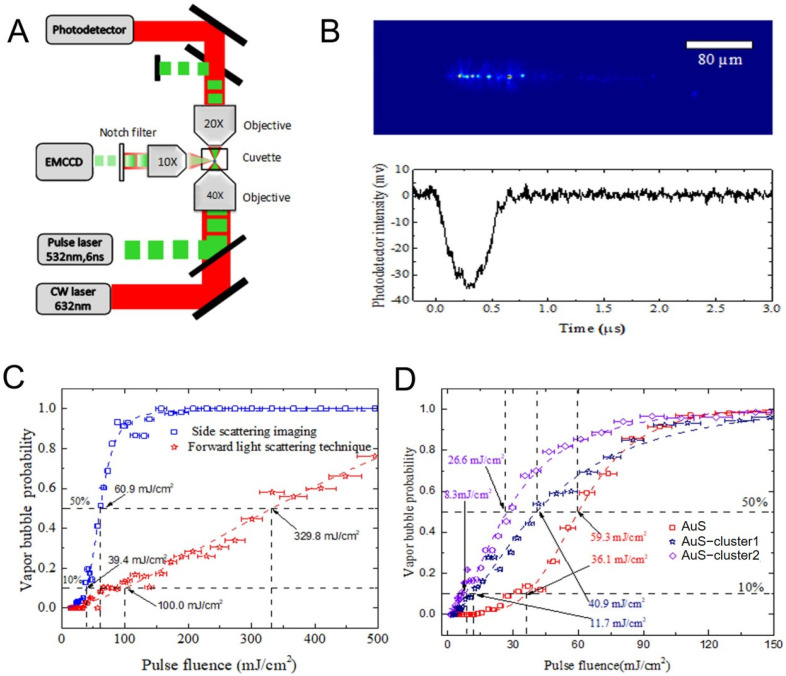
The fluence threshold measurement of pulsed laser induced vapor bubble mediated by PEG-AuS. Notes: (**A**) The schematic diagram of the experimental setup for vapor bubble measurement. (**B**) Side-scattering image of vapor bubble captured by camera (top) and the time response of intensity of the forward probe beam for vapor bubble (bottom). (**C**) The probability of vapor bubble detected by side-scattering imaging and forward light scattering detection. (**D**) The probability of vapor bubble formation of PEG-AuS and AuS cluster measured by side-scattering imaging technique.

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
