# Peer review of "Influence of Parameters on the Death Pathway of Gastric Cells Induced by Gold Nanosphere Mediated Phototherapy"

_nanomaterials, 2022, doi:10.3390/nano12040646_

Round 1

Reviewer 1 Report

The authors have performed a detailed study of the effect on the concentration of gold nanoparticles, incubation time, light doses on the cyto and fototoxicity in two different line of gastric cancer cells (SGC-7901 and GES-1) under a nanosecond laser irradiation. They have studied the main mechanism of action to kill cells (PPT,  ROS production or vapor bubbles formation) of these AuS nanoparticles under different conditions. The have also studied the cell pathway (necrosis and apoptosis) and the cellular uptake processes.

In general, the article is well written. The experimental section is well detailed and the performed experiments are also well designed. However, it was difficult for me to evaluate the results since in the manuscript, since all the figures and supporting section were missing in the version I have be given. Please, to properly revised this work, let me see those date.

Nevertheses, and after including those figures  I consider a suitable study to be published in nanomaterials.

 I also have some minor questions:

-Section 3.2 at 0.0537 nM and any incoubation time (2.4 and 6.h) the nanoparticles are not toxic. However, when the concentration is increase to 0.1075 nM the viability has descends to 86%, 66% and 58% for 2, 4, and 6 h of incubation time. The authors set the limit at this concentration at 4h, which the nanoparticles causes already a noticeable toxicity. Most surprisingly, they set the upper limit at 0.215 nM concentration and 6h of incubation, which under my opinion these conditions should cause a great toxicity if we take into account that at 0.1075 nM and 4 h of incubation there is a 44% of mortality. Therefore, it is difficult to believe that 0.215 nM and 6h of incubation is a “Secure” range.

-section 3.3. and 3.5. The concentration of AuS used for the experiments should be added in the main text

Author Response

Dear Editors and Reviewer,

Thank you very much for your evaluation and comments from the reviewer for our manuscript. We have learned carefully from the editor’s and reviewer’s comments, which are very valuable and very helpful for revising and improving our paper. After studying the critical comments, we have responded point by point and made corresponding changes in our manuscript. Our responses to the editor’s and reviewer’s comments are as follows:

Point 1: It was difficult for me to evaluate the results since in the manuscript, since all the figures and supporting section were missing in the version I have be given. Please, to properly revised this work, let me see those date.

Response 1: We are very sorry to provide the manuscript version with missing figures and supporting section. We have submitted our revised manuscript.

Point 2: Section 3.2 at 0.0537 nM and any incoubation time (2.4 and 6.h) the nanoparticles are not toxic. However, when the concentration is increase to 0.1075 nM the viability has descends to 86%, 66% and 58% for 2, 4, and 6 h of incubation time. The authors set the limit at this concentration at 4h, which the nanoparticles causes already a noticeable toxicity. Most surprisingly, they set the upper limit at 0.215 nM concentration and 6h of incubation, which under my opinion these conditions should cause a great toxicity if we take into account that at 0.1075 nM and 4 h of incubation there is a 44% of mortality. Therefore, it is difficult to believe that 0.215 nM and 6h of incubation is a “Secure” range.

Response 2: We are very sorry that we didn’t describe these results in section 3.2 clearly. As you can see, at 0.0537 nM and any incubation time (2. 4 and 6 h) the nanoparticles are not toxic. And when the concentration is increase to 0.1075 nM the viability on SGC-7901 cells (gastric cancer cell line) has descends to 86%, 66% and 58% for 2, 4, and 6 h of incubation time. However, we found that the viability on GES-1 (immortalized normal gastric mucosal cell line) with the same concentration and incubation time was almost no decrease. The secure concentration is 0.215nM. Based on the principle of killing tumor cells to the greatest extent while having little toxic on normal cells, we suggested that the used concentration of PEG-AuSs for drug delivery and imaging diagnose should under 0.0537 nM for 6 h as much as possible and no more than 0.1075 nM for 4 h, and the concentration can be increased to 0.215 nM for 6 h and even to 0.43 nM for 2 h to gastric cancer therapy. We have revised this part as:“To determine the initial value of PEG-AuS induced cytotoxicity on SGC-7901 gastric cancer cell line and GES-1 immortalized normal gastric mucosal cell line, CCK-8 cell viability analysis was performed. The incubation time and the dose of PEG-AuS are essential factors that influence on the cytotoxicity. Hence, the cell viability of PEG-AuS was evaluated with the incubation time or the dose changed. As shown in Figure 3, 0.0537 nM PEG-AuS induce little cytotoxicity on SGC-7901 cells, despite the incubation time being 2, 4 and 6 h. Increased the dose to 0.1075 nM, the cell viability at 2, 4 and 6 h was decreased to 86%, 66%, and 58% respectively. Furthermore, when the dose of PEG-AuS or the incubation time was increased, the cell viability decreased accordingly. Increased the dose to 0.43 nM, the cell viability was nearly inhibited at 6 h. And increased the dose to 1.72 nM, the cell viability was completely inhibited even at 2h. Compared with SGC-7901 cells, the little cytotoxicity of PEG-AuSs on GES-1 cells was below 0.215 nM. Above 0.43 nM, the cell viability was down quickly, regardless of the incubation time (2, 4 and 6 h). Generally, the incubation time and the drug dose had a higher influence on SGC-7901 cells compared with normal GES-1 cells. This may be caused by its higher intracellular uptake efficiency related with proliferation ability. Hence, the used concentration of PEG-AuSs for drug delivery and imaging diagnose should under 0.0537 nM for 6 h as much as possible and no more than 0.1075 nM for 4 h, and based on the principle of killing tumor cells to the greatest extent while having little toxic on normal cells to gastric cancer therapy, the concentration can be increased to 0.215 nM for 6 h and even to 0.43 nM for 2 h.”

Point 3: section 3.3. and 3.5. The concentration of AuS used for the experiments should be added in the main text.

Response 3: Thanks very much for your comments. We have added the concentration of AuS used for the experiments in section 3.3 and 3.5.

Reviewer 2 Report

Judging by the text of the manuscript, the authors' study is devoted to solving a very urgent problem, namely, determining the threshold values of the dose of laser radiation for effective exposure on cancer cells in the presence of gold plasmon nanospheres. The authors describe in detail the logic of planning and conducting experiments, processing results. The results of the conducted research are definitely of high practical significance.

The main drawback of the manuscript is the absence of Figures 1-10 in the file, which are nevertheless referenced in the text. Perhaps the "loss" of Figures is a kind of casus. But this is an obstacle to a good assessment of the manuscript.

There are also some methodological errors. So, for example, in subsection 3.8, the authors give the value of the threshold dose with an accuracy of 0.02%. At the same time, there is no initial information for calculating this quantity - the intensity, dimensions and structure (distribution over the cross section) of the laser beam, the actual duration of the "nanosecond pulse". In this case, the temperature increase is given only in units of degrees. A certain discrepancy between the levels of measurement errors (determination) of various parameters is obvious. Therefore, the accuracy of determining the threshold exposure doses given in Section "4. Conclusion" does not seem to be fully justified.

Author Response

Dear Editors and Reviewer,

Thank you very much for your evaluation and comments from the reviewer for our manuscript. We have learned carefully from the editor’s and reviewer’s comments, which are very valuable and very helpful for revising and improving our paper. After studying the critical comments, we have responded point by point and made corresponding changes in our manuscript. Our responses to the editor’s and reviewer’s comments are as follows:

Point 1: The main drawback of the manuscript is the absence of Figures 1-10 in the file, which are nevertheless referenced in the text. Perhaps the "loss" of Figures is a kind of casus. But this is an obstacle to a good assessment of the manuscript.

Response 1: Sorry for this mistake. We have added the Figures in the text.

Point 2: There are also some methodological errors. So, for example, in subsection 3.8, the authors give the value of the threshold dose with an accuracy of 0.02%. At the same time, there is no initial information for calculating this quantity - the intensity, dimensions, and structure (distribution over the cross section) of the laser beam, the actual duration of the "nanosecond pulse". In this case, the temperature increase is given only in units of degrees. A certain discrepancy between the levels of measurement errors (determination) of various parameters is obvious. Therefore, the accuracy of determining the threshold exposure doses given in Section "4. Conclusion" does not seem to be fully justified.

Response 2: Thanks for your comments and sorry for these mistakes. We have added the initial information for calculating the radiant dose in this study in subsection 2.5. The radiant intensity of nanosecond pulsed laser is about 1.5, 3, 6,12, 24 mJ, the actual duration of the nanosecond pulsed laser is 0.1s, the radius of laser beam is 0.5cm. Hence, the used doses in subsection 3.8 are about 11.47 mJ/cm2 and 22.93 mJ/cm2. There are inevitable measurement errors in measuring laser intensity. To avoid this problem to the greatest extent, all of experiments in this study have been repeated several times, and then three experiments with the same radiant intensity as the initial intensity will be selected and analyzed. Hence, the accuracy of determining the threshold exposure doses could be ensured. In order to present the conclusion clearly, the “4.Conclusion” have revised as: “In this study, we revealed the main parameters and threshold values for PEG-AuS-mediated gastric cancer phototherapy with nanosecond pulsed laser irradiation, evaluated the pathway of induced cell death under the conditions revealed above and discussed the roles of PTT,photochemical and vapor effects which can induce the cell death. The results showed that the non-cytotoxic concentration of PEG-AuS in gastric cancer cells is 0.053 nM, therefore the used concentration of PEG-AuSs for drug delivery and imaging diagnose should under 0.0537 nM for 6 h as much as possible. In addition, based on the principle of killing tumor cells to the greatest extent while having little toxic on normal cells to gastric cancer therapy, the concentration can be increased to 0.1075 even to 0.215 nM if which is used to phototherapy, because of 0.215 nM is non-cytotoxic concentration of PEG-AuS in normal gastric mucosal cells. Treated with 0.1075 nM PEG-AuS for 4 h, 22.93 mJ/cm2 radiation energy (6mJ radiant intensity, 3000 pulses, 0.5cm radiation radius) or 6h, 11.47 mJ/cm2 radiation energy (3mJ radiant intensity, 3000 pulses, 0.5cm radiation radius), the anti-growth effect could be exhibited significantly. And treated with 0.1075 nM PEG-AuS for 4 h, 22.93 mJ/cm2 radiation energy, apoptosis can be induced in gastric cancer cells and the SOG-mediated photochemical effect induced by AuS is better than its photothermal and vapor effects. After prolonging the treated time to 6 h, even reducing the irradiation energy to 11.47 mJ/cm2, cell death could be still induced in the gastric cancer cells and the pathway of cell death mainly depended on necrosis. Under these conditions, the ROS-mediated photochemical effect induced by AuS is better than its photothermal and vapor effects. And the threshold of irradiation dosage to induce vapor effect could be decreased by AuS aggregation. This revealed that PEG-AuS could induce cell death quickly through ROS-mediated photochemical effect and vapor effect with low irradiation by prolonging the binding time of AuS. Generally, PEG-AuS under nanosecond pulsed laser radiation is a safe and effective agent and it is suitable for phototherapy and drug delivery system for gastric cancer therapy.”

Reviewer 3 Report

The article entitled “Influence of parameters on the death pathway of gastric cells induced by gold nanosphere mediated phototherapy” present interesting approach, however, it is difficult to follow the veracity of the results without graphical representation. Therefore, the article should be improved in this context.

Author Response

Dear Editors and Reviewer,

  Thank you very much for your evaluation and comments from the reviewer for our manuscript. We have learned carefully from the editor’s and reviewer’s comments, which are very valuable and very helpful for revising and improving our paper. After studying the critical comments, we have responded point by point and made corresponding changes in our manuscript. Our responses to the editor’s and reviewer’s comments are as follows:

Point 1: The article entitled “Influence of parameters on the death pathway of gastric cells induced by gold nanosphere mediated phototherapy” present interesting approach, however, it is difficult to follow the veracity of the results without graphical representation. Therefore, the article should be improved in this context.

Response 1: We are very sorry to provide the manuscript version with missing figures. We have submitted our revised manuscript.

Round 2

Reviewer 1 Report

The authors have satisfactory replied to my comments. The manuscript has been improved and the data missing included

It is suitable for publication in nanomaterials in the present form

Reviewer 2 Report

The results presented in the manuscript under review are of significant scientific and practical value. The extensive experimental studies carried out by the authors and their analysis are a good example of how to choose the most effective methods of treating gastric cancer based on a comparative assessment of various physicochemical factors.

Pegylated gold nanospheres, energy of laser impulses of nanosecond duration, PEG-AuS incubation time, type of cages (normal gastric tissue cells, gastric cancer cells), as concentration of the agent PEG-AuS act as such various factors of influence.

At the same time, cell cytotoxicity with respect to PEG-AuS, the degree of photothermal exposure, the conditions for initiating necrosis and apoptosis, photochemical effects, and the generation of vapor bubbles are studied.

At the same time, cell cytotoxicity to PEG-AuS, the degree of photothermal exposure, the conditions for initiating necrosis and apoptosis, photochemical effects, and the generation of vapor bubbles are investigated.

Despite the fact that the results were obtained under quite specific conditions (the same duration of laser pulses, irradiation time, size of nanoparticles), the approaches demonstrated by the authors in assessing various criteria and choosing a gold nanosphere mediated phototherapy strategy seem to be very useful.

As minor remarks, there are no scale marks on Fig. 7 and Fig. 8. Also, the temperature color scale on Fig. 8. unreasonably expanded, as a result, the temperature gradation on the temperature map becomes poorly discernible.

Reviewer 3 Report

thank you very much for the providing of the prompt answer. The article now is ready for publication.